# Kinetic Analysis and Epitope Mapping of Monoclonal Antibodies to *Salmonella* Typhimurium Flagellin Using a Surface Plasmon Resonance Biosensor

**DOI:** 10.3390/antib8010022

**Published:** 2019-03-06

**Authors:** Devendra Bhandari, Fur-Chi Chen, Shreya Hamal, Roger C. Bridgman

**Affiliations:** 1Department of Agricultural and Environmental Sciences, Tennessee State University, Nashville, TN 37209, USA; dbhandar@my.tnstate.edu (D.B.); ssinghha@my.tnstate.edu (S.H.); 2Department of Human Sciences, Tennessee State University, Nashville, TN 37209, USA; 3Hybridoma Facility, Auburn University, Auburn, AL 36830, USA; bridgcr@auburn.edu

**Keywords:** *Salmonella* Typhimurium, flagellin, epitope mapping, kinetic analysis, surface plasmon resonance

## Abstract

*Salmonella* Typhimurium is one of the leading causes of foodborne diseases worldwide. Biosensors and immunoassays utilizing monoclonal antibodies are widely used for the detection and subtyping of *S.* Typhimurium. However, due to insufficient information on the nature of binding with *S.* Typhimurium flagellin, the selection of appropriate antibodies for assay development is a cumbersome task. Hence, we aimed to compare the binding kinetics of a panel of monoclonal antibodies and their relative binding sites to flagellin antigen using a surface plasmon resonance biosensor. Initially, the flagellin was captured on the sensor surface through an immobilized anti-flagellin antibody. The interactions of different concentrations of monoclonal antibodies to flagellin were determined, and binding curves were fitted using 1:1 bio-interaction model to calculate the kinetic parameters. For epitope mapping, pairwise comparisons were completed to determine the binding inhibition of each paired combination of monoclonal antibodies. It was found that these monoclonal antibodies differed significantly (*p* < 0.05) in association rate, dissociation rate, and equilibrium dissociation constants. Of the five monoclonal antibodies, only two interfered with the binding of each other. Four distinct epitopes located within a 23 kDa domain of flagellin were identified. Findings from this study provide crucial information needed for the further development and optimization of biosensors and other immunoassays for the detection and subtyping of *Salmonella*.

## 1. Introduction

*Salmonella* Typhimurium is one of the leading cause of foodborne diseases worldwide [1]. The bacterium has 6–10 flagella that are arranged peritrichously around the cell for its mobility [2]. A single flagellum is divided into three distinct sub-structures: Hook, basal body, and filament [3]. The filament, which extends to the extracellular space, is up to 10 µm long and is composed of approximately 20,000 subunits of a single species of protein called flagellin. *S.* Typhimurium has two non-allelic genes that encode two antigenically distinct flagellins. The *fliC* gene encodes phase 1 flagellin and the *fljB* gene encodes phase 2 flagellin [4]. Individual cells produce either phase 1 or phase 2 flagellin by a mechanism called phase variation [5]. Serotype Typhimurium have ‘i’ as phase 1 flagellin and ‘1,2’ as phase 2 flagellin [6], which are identical for the first 71 amino acids and the last 46 amino acids and are antigenically distinct for surface exposed amino acids in the middle portion [7]. Both flagellins have mostly hypervariable regions spanning from 190–291 amino acids where most antibodies bind. The binding of monoclonal antibodies with either ‘i’ flagellin or ‘1,2’ flagellin is determined by the specificity of the antibody to ‘i’ and/or ‘1,2’ flagellins [8].

In recent years, the production of high-quality antibodies in larger quantities has become easier, especially after the advent of hybridoma technology [9,10]. Hybridoma technology has been used extensively to produce different types of antibodies for therapeutic, disease diagnostic, and research purposes [9,10,11,12,13,14]. Most importantly, various antibodies against *Salmonella* flagellin were also produced utilizing hybridoma technology [15,16,17,18,19]. The availability of different anti-flagellin antibodies along with the easy extraction of *Salmonella* flagellin [20] facilitates the development of numerous immunoassays like enzyme-linked immunosorbent assays (ELISA) and Surface Plasmon Resonance (SPR) assays. ELISA detects and quantifies *Salmonella* flagellin using a labeled antibody. A number of ELISA have been successfully used for the detection of *Salmonella* from food and environmental samples [21,22,23,24]. In addition to ELISA, SPR assays have recently been applied for the detection of *Salmonella* [25,26,27,28]. Similarly, SPR assays developed with numerous anti-flagellin antibodies from various strains of *Salmonella* have a potential for subtyping *Salmonella* based on antibody and flagellin binding kinetics [29]. 

Despite a multitude of applications for anti-flagellin antibodies, the sensitivity of the assays did not improve as expected. The sensitivity and reliability of the assays can be enhanced by using well-characterized antibodies in terms of the binding epitope and binding kinetics. Information about the epitope map facilitates the selection of one or more antibodies based on the objective of the assay to be developed. Especially in the sandwich type assay performed in ELISA and SPR, two antibodies should have different binding sites on the antigen. If the antibodies have the same binding epitope, the detection antibody would fail to give a signal. Similarly, binding kinetics play a vital role in the sensitivity and specificity of the developed immunoassays. The association rate shows how fast antigens bind to an antibody, whereas the dissociation rate shows how fast the binding disrupts. More importantly, affinity, which is inversely related to the equilibrium dissociation constant, indicates how much antibody-antigen complex is formed at the equilibrium. Knowledge of association rate, dissociation rate, and affinity between flagellin and anti-flagellin antibodies benefits the selection of appropriate antibodies in assay development. 

The SPR optical biosensor works on the principle of total internal reflection and measures the antibody-antigen interaction in terms of change in the reflective angle [30]. SPR assays are fast, label-free, and allow real-time monitoring of the biomolecular interactions with higher sensitivity [31]. In contrast to covalent immobilization, the capture-based antibody immobilization on the SPR sensor chips allows for improved orientation, and thus the reactivity potential of the antibody is increased as compared with most ELISA, where the antibody is physically adsorbed on the microplate [32]. Due to these advantages, SPR provides better epitope mapping alternatives over commonly used ELISA methods. SPR has been extensively used for epitope mapping of different antibodies [33,34,35,36,37]. Similarly, kinetic analyses of different antibodies with their respective antigens were successfully accomplished with SPR biosensors [38,39,40,41,42,43,44,45]. 

In this study, we have developed an SPR assay using the capture-based immobilization to determine the relative binding sites of five monoclonal antibodies to *S.* Typhimurium flagellin. Additionally, we have investigated the binding kinetics of these monoclonal antibodies using the same approach.

## 2. Materials and Methods

### 2.1. Materials and Equipment

Salmonella Typhimurium (ATCC 13311) was purchased from American Type Culture Collection (Manassas, VA, USA), and stored at −80 °C before use. 

Tryptic soy agar (TSA) and tryptic soy broth (TSB) were supplied by Remel, Thermo Fisher Scientific Inc. (Lenexa, KS). 10× phosphate buffered saline (PBS), Tween 20, 4× XT sample buffer, 20× XT reducing agent, Trans-Blot® Turbo™ RTA Mini Nitrocellulose Transfer Kit, Precision Plus Protein™ WesternC™ Standard, Precision Protein^TM^ Strep Tactin-HRP conjugate, Goat Anti-Mouse HRP conjugate and Clarity Western ECL substrate were acquired from Bio-Rad Laboratories, Inc. (Hercules, CA, USA). Bovine serum albumin (BSA) was bought from Fisher Scientific (Hampton, NH). Lyophilized trypsin powder was purchased from Sigma-Aldrich Co. (St. Louis, MO). Bolt™ 4–12% Bis-Tris Plus gels were purchased from Life Technologies (Carlsbad, CA). The 1× PBS with 0.05% Tween 20 (PBST) was used as a working buffer. Primary and secondary antibodies were diluted in PBST containing 1% BSA.

N-(3-dimethylaminopropyl)-N′-ethylcarbodiimide hydrochloride (EDC), N-hydroxysuccinimide (NHS), ethanolamine hydrochloride, and sodium acetate were acquired from Sigma-Aldrich Inc (St. Louis, MO, USA). All solutions were prepared using distilled, deionized, filtered and degassed water. 

SPR assays were performed using a Reichert SR7500 DC biosensor and its associated software called Integrated SPR Autolink Version 1.1.14-T (Reichert Technologies, Buffalo, NY, USA). TraceDrawer Version 1.6.1 by Ridgeview Instruments AB (Upsala, Sweden) was used to process and analyze SPR data. The 500 kDa Carboxymethyl Dextran Hydrogel Surface Sensor Chip (SR7000 gold sensor slide) was purchased from Reichert Inc, NY, USA.

### 2.2. Preparation of Salmonella Typhimurium Flagellin

Cultures of S. Typhimurium were prepared on TSA plates by incubating at 35 °C for 24 h after retrieving from the freezer. The cultrues were further propagated by transferring to additional TSA plates. Cells from each plate after incubating at 35 °C for 24 h were collected by washing with 1 mL PBS. The recovered suspensions were centrifuged (3000× *g*, 10 min) and the cell pellets were collected. A volume of 10 mL of 250 mM Glycine-HCl, pH 2.0, was added to the pellets and then vortexed. After incubating the suspension for 30 min at room temperature, centrifugation was done to collect the supernatant. The pH of the supernant was adjusted to 7.0, and an Amicon® Ultra-15, 10K centrifugal filter was used to concentrate and to exchange buffer to PBS. Finally, the sample from the filter was collected and the volume was adjusted to 500 μL using PBS. The flagellin preparations were analyzed using Bolt™ 4–12% Bis-Tris Plus gels to check for purity. Presence of minor fragments in the flagellin preparation was noticed and described in Section 3.1. The flagellin preparations were stored at −80 °C and later retrieved for antibody production. 

### 2.3. Production of Anti-Flagellin Antibodies 

Flagellin preparation (1.0 mg/mL) in PBS containing 0.3% of sodium dodecyl sulfate was heated in boiling water for 10 min. Four BALB/c mice (7–10 weeks old) were immunized either subcutaneously or intraperitoneally with 0.15 mg of the heat-treated flagellin mixed 1:1 (*v*/*v*) with Freund’s complete adjuvant followed by two booster injections at 4-week intervals with 0.1 mg per mouse of flagellin mixed 1:1 (*v*/*v*) with Freund’s incomplete adjuvant. Test sera were collected by tail bleeding 10 days after each injection; the titer of the sera against the flagellin preparation was then determined by indirect ELISA. The mouse exhibiting the highest serum titer to flagellin then received a final boost of 0.1 mg of the flagellin in PBS four days before the fusion. Spleen cells from the selected mouse were fused with the myeloma cell line (P3363.Ag8.653., ATCC CRL 1580) at a ratio of 5:1 in the presence of polyethylene glycol (molecular weight, 4000). The hybridoma cells were subsequently diluted to an appropriate density and cultured in hypoxanthine-aminopterinthymidine medium.

The medium was changed twice to remove residual antibodies before the initial screening against flagellin using indirect ELISA. For secondary selection, the positive cells from the initial screening were transferred to larger wells and cultured for three more days. Western blots were performed to confirm the binding to flagellin at 50 kDa. The selected cell lines were cloned at least twice by a limiting dilution method and subsequently maintained in liquid nitrogen. Ascites fluids containing the antibodies were obtained from Pristane primed mice 10–14 days after intraperitoneal injection of the hybridoma cells. Antibodies were separated from the ascites fluid using a Protein A Cartridge with MAPS II buffer (Bio-Rad). The purified antibodies were dialyzed against PBS overnight at 4 °C with several changes of dialysis buffer. The concentrations of antibodies in the final preparations were determined by UV absorption at 280 nm. The purified antibodies were stored in aliquote at −20 °C with addition 0.05% of sodium azide. A panel of five monoclonal antibodies (MAbs 1C8, 1E10, 3H8, 5F11, and 7E3) was produced. MAbs 1C8, 1E10, and 5F11 belong to IgG_1_; MAb 7E3 and MAb 3H8 belong to IgG_2a_ and IgG_2b_, respectively, as determined by the Mouse Isotyping Kit (Bio-Rad). 

### 2.4. Western Blot Assay

Flagellin preparation was mixed with tripsin at a protein ratio of 10:1 (*w*/*w*) and incubated for 2, 4, 8, and 16 min at room temperature. Immediately after incubation for the specified time, the tripsin-treated flagellin was mixed with an equal volume of 2× XT loading buffer and heated at 95 °C for 5 min. An untreated flagellin sample was also prepared with an equal volume of 2× XT loading buffer in a similar way. Precision Plus Protein™ WesternC™ Standards were used for the molecular weight calibrations. Flagellin samples (untreated and tripsin-treated) and standards were separated using Bolt™ 4–12% Bis-Tris Plus gels. Western blot was carried out on the proteins separated by gel electrophoresis and subsequently transferred to a nitrocellulose membrane. The membrane was blocked overnight with 3% nonfat dry milk powder in PBST and consecutively incubated with a monoclonal antibody (0.4 µg/mL in PBST, 1% BSA) and a goat anti-mouse IgG conjugated to horseradish peroxidase (0.2 µg/mL in PBST, 1% BSA). Binding activity was finally detected with Clarity Western ECL substrate prepared according to the manufature’s instruction. The image of the blot was generated by the ChemiDoc MP system and analysis of band patterns was completed by using ImageLab software (Bio-Rad, CA, USA).

### 2.5. Immobilization of the SPR Sensor Surface

The 500 kDa Carboxymethyl Dextran Hydrogel Surface Sensor Chip (SR7000 gold sensor slide) was installed onto a Reichert SR7500DC biosensor following the manufacturer’s instruction. The sensor surface was then preconditioned by running PBST at 20 µL/min until a stable baseline was obtained. The flow rate of 20 µL/min and temperature of 25 °C were maintained throughout the immobilization process. In order to activate carboxy groups on the surface of the sensor chip, a fresh preparation of 40 mg EDC and 10 mg NHS disolved in 1 mL water was injected onto the sansor surface for 5 min. To the activated surface, MAb 1E10 diluted in 10 mM sodium acetate (150 µg/mL), pH 5.2, was injected only to the left channel of the surface for 5 min. Then, BSA dissolved in 10 mM sodium acetate (75 µg/mL) was injected to both channels to saturate the remaining active sites. Finally, quenching solution (1.0 M ethanolamine, pH 8.5) was injected for 5 min to deactivate carboxyl groups and to wash away the unbound antibody and BSA. A continuous flow of runing buffer (PBST) at 20 µL/min was maintained after the completion of antibody immobilization. SPR assays were carried out after a stable baseline was achieved. 

### 2.6. Kinetic Analysis Using SPR

Kinetic analysis was initiated by injection of flagellin (17 µg/mL in PBST) onto the MAb 1E10 immobilized sensor surface for 4 min, followed by injection of PBST for 6 min. The binding kinetics of an individual monoclonal antibody to the flagellin captured on the sensor surface was determined from serial dilutions (four different concentrations) of the antibody under study. Each dilution was injected for 4 min (association) followed by PBST for 6 min (dissociation). After dissociation, the flagellin-antibody complex was removed from the immobilized surface by injecting regeneration buffer (10 mM Glycine-HCl, pH 3.0) for 4 min followed by PBST for 6 min. The same procedures were repeated for the remaining three dilutions of the antibody. After the analyses were completed, signals from left channel were subtracted from signals from their respective reference channel (the right channel). Additionally, the resulting curves were further corrected with a blank signal from the PBST. The SPR responses from four concentrations of the same antibody were fitted to a 1:1 bio-interaction model (Langmuir fit model) utilizing TraceDrawer Software. Association rate constant (k_a_), dissociation rate constant (k_d_), and maximum binding (Bmax) were fitted globally, whereas the BI signal was fitted locally. The equilibrium dissociation constant (K_D_) was calculated from the ratio of k_d_/k_a_.

### 2.7. Pairwise Epitope Mapping Using SPR

Twenty-five paired combinations from five monoclonal antibodies were used to study the binding inhibition on one antibody due to the other. Initially, flagellin (17 µg/mL in PBST) was introduced onto the MAb 1E10 immobilized sensor surface for 4 min followed by PBST for 4 min. A pair of antibodies (11 µg/mL in PBST) were prepared and injected consecutively. The first antibody was injected for 4 min followed by 4 min of dissociation, and immediately the second antibody was injected for 4 min, followed by 4 min of dissociation. Finally, regeneration of the surface was performed by injecting 10 mM Glycine-HCl, pH 3.0, for 4 min to remove the bound antigen-antibodies complex and to prepare the surface for the next analysis. Experiments were conducted with replicated samples for each pair of monoclonal antibodies. In order to quantify the interference of one antibody binding to another, we compared each of the pairs using the binding ratio as described in a previous report [37]. The binding ratio was defined as the Bmax of the second antibody divided by the Bmax of the first antibody. 

### 2.8. Data Analysis

Averages and standard deviations of data from SPR analyses were calculated. Analysis of variance (ANOVA) and multiple comparisons were performed with IBM SPSS Statistics 24 software (Armonk, NY, USA). Tukey’s post hoc tests were performed for multiple comparisons.

## 3. Results

### 3.1. Limited Proteolysis and Western Blot Analysis

The results indicated that the undigested flagellin preparations, in addition to the intact flagellin (50 kDa), contain minor fragments in the range between 33 and 41 kDa. The presence of flagellin epitopes on these fragments was evident when different monoclonal antibodies were tested. In summary, five anti-flagellin antibodies produced three different banding patterns with undigested flagellin (0 min, Figure 1). MAb 1C8 and MAb 7E3 produced similar banding patterns with one major band at 50 kDa and two minor bands at 38 kDa and 33 kDa. Another type of banding pattern, produced by MAb 3H8 and MAb 5F11, had two bands at 50 kDa and 38 kDa. Interestingly, MAb 1E10 produced a unique type of banding pattern. As compared to MAbs 1C8, 7E3, 3H8, and 5F11, MAb 1E10 identified a distinct band at 41 kDa. Western blot analysis showed that all five monoclonal antibodies detect at least two bands at 50 kDa and 38 kDa. 

Proteolytic fragments of flagellin were produced by limited trypsin digestion to compare the different banding patterns of five monoclonal antibodies. Based on the similarity of the banding patterns, these antibodies can be categorized into two groups. MAb 1C8 and MAb 7E3 had similar banding patterns, whereas MAb 3H8, MAb 5F11, and MAb 1E10 had similar band patterns. All five antibodies detected flagellin fragments at 38, 33, and 23 kDa. The results from the Western blot of the trypsin-digested fragments suggested that all epitopes recognized by these five antibodies are collocated within the smallest fragment (23 kDa). MAb 1C8 and MAb 7E3 detected two additional bands at 27 and 25 kDa. Western blot also revealed that there was only one major band at 33 kDa for MAb 1C8 and MAb 7E3, but for the other three antibodies, there were two major bands at 38 and 33 kDa.

The results of the Western blot analysis suggested that there were at least three distinct epitopes of flagellin which the five antibodies can react with. The fact that MAb 1E10 was able to bind to a unique size of a flagellin fragment at 41 kDa suggested that the epitope for 1E10 is different from the rest of the antibodies. Based on this observation, MAb 1E10 was selected as the capturing antibody for the flagellin in the kinetic analysis and epitope mapping. 

### 3.2. The Binding Kinetics of Monoclonal Antibodies to Flagellin 

A typical sensorgram of the interactions of four different concentrations of the same antibody to the flagellin captured by MAb 1E10 immobilized on the sensor surface is presented in Figure 2. SPR curves were fitted to a 1:1 interaction model. Finally, the association rate constant (*k_a_*), dissociation rate constant (*k_d_*), and the equilibrium dissociation constant (*K*_D_) of the antibody were determined. The kinetic data of four monoclonal antibodies are summarized in Table 1.

The flagellin was first captured by MAb 1E10 immobilized on the sensor surface, and four concentrations of the same antibody were analyzed consecutively. The 1:1 interaction model was used, and the association rate constant (*k_a_*), dissociation rate constant (*k_d_*), and the equilibrium dissociation constant (*K*_D_) were calculated. Values in parentheses are standard deviations. ANOVA was performed to determine the significant difference. A Tukey’s post-hoc test was performed for multiple comparisons, and numbers followed by different letters were significantly different. 

It was found that four antibodies differ significantly (*p* < 0.05) for their association rate (*k_a_*), dissociation rate (*k_d_*), and the equilibrium dissociation constant (*K*_D_) based on their interactions with the flagellin captured by MAb 1E10 (Table 1). MAb 5F11 had a faster association rate and was 2.4, 3.2, and 2.3-fold higher than MAb 1C8, MAb 7E3, and MAb 3H8, respectively. MAb 1C8, MAb 7E3, and MAb 3H8 showed no significant difference in their association rates. 

The dissociation rate (*k_d_*) and equilibrium dissociation constant (*K*_D_) of MAb 3H8 were significantly different from that of MAb 5F11. However, the *k_d_* and *K*_D_ values of MAb 1C8 and MAb 7E3 were not significantly different. MAb 3H8 had a faster dissociation, which was 3.1, 2.1, and 5.4-fold higher than MAb 1C8, MAb 7E3, and MAb 5F11, respectively. As a result, MAb 3H8 had the lowest affinity as indicated by the highest *K*_D_ with flagellin, and its value was 2.9, 1.5, and 12.3-fold higher than that of MAb 1C8, MAb 7E3, and MAb 5F11, respectively. 

### 3.3. Epitope Mapping by SPR Inhibition Assay 

The relative positions of flagellin epitopes between a pair of monoclonal antibodies were studied using the SPR inhibition assay which measures the competitive binding of one antibody due to the previous binding of another antibody. Typical examples of the inhibition/non-inhibition of the binding of an antibody to another are illustrated in Figure 3. At first, flagellin was captured by MAb 1E10 immobilized on the sensor surface. The average binding response of flagellin to the immobilized MAb 1E10 was 137 micro-Refractive Index Units (µRIU). Immediately after, MAb 1C8 was injected and MAb 3H8 was followed. MAb 3H8 bound with the flagellin and yielded a comparable SPR signal to MAb 1C8 (Bmax 250 and 255 µRIU, respectively). It indicated that MAb 1C8 has no inhibition on the binding of MAb 3H8. In a completely opposite case, after MAb 1C8, MAb 7E3 was injected. MAb 7E3 did not bind to the flagellin (0 µRIU). It indicated that the binding of MAb 7E3 was completely inhibited by MAb 1C8.

The inhibition of the binding was calculated as the binding ratio by dividing the Bmax of the second antibody with the Bmax of the first antibody. A summary of the binding ratio of paired monoclonal antibodies on the immobilized flagellin is presented in Table 2. Antibody binding ratios of approximately <1, = 1 and >1 were expected for binding inhibition, no binding inhibition, and binding enhancement, respectively. The binding ratios revealed that the same antibody injected consecutively one after another did not produce a binding signal during the second injection except in the case of two antibodies. MAb 3H8 and MAb 5F11 produced approximately 20% and 21% binding signals in the second injection as compared to their first injection. 

Interestingly, it was found that antibody pairs (MAb 1C8 and MAb 7E3) interfered with the binding of each other. The data suggested that MAb 7E3 and MAb 1C8 have an identical epitope or overlapping epitopes. The binding ratio of MAb 1C8 after binding of MAb 7E3 was 0.12, whereas the ratio of MAb 7E3 after binding of MAb 1C8 was 0.29. In other words, the binding of MAb 7E3 had reduced the binding of MAb 1C8 by 88%, and in contrast, the binding of MAb 1C8 had reduced the binding of MAb 7E3 by 71%. 

On the other hand, pairs of monoclonal antibodies that have a binding ratio of equal to or more than 1.0 indicate either no inhibition or enhancement of binding by the previously injected antibody. There are several cases where enhancement of the binding of the secondary antibody was observed. This may be attributed to the differences of the affinities between the two antibodies and the conformational changes of flagellin caused by the binding of the first antibody [46]. Collectively, it was clear that the first antibody did not hinder the binding of the second antibody except in the case of MAb 1C8 and MAb 7E3.

There was no binding response (0 µRIU) when the same MAb 1E10 was injected after flagellin was captured by the immobilized MAb 1E10. It suggested that the binding of 1E10 to flagellin was univalent and all binding sites of MAb 1E10 were fully occupied by the immobilized MAb 1E10 (the capturing antibody). However, antibodies injected after MAb 1E10 produced signals with no inhibition. It was clear that MAb 1E10 did not hinder the binding of other antibodies when injected consequently.

## 4. Discussion

Western blot analysis showed that five monoclonal antibodies detected three types of banding patterns with undigested flagellin. After limited trypsin digestion, two types of banding patterns were observed. The presence of minor fragments in the undigested flagellin was noticed in the range between 38–41 kDa. Smaller fragments (41, 42 kDa) of *S.* Typhimurium flagellin due to the deletion of parts of the outer domain have been reported previously [47]. 

Of the five monoclonal antibodies, only MAb 1E10 was able to detect the 41 kDa fragments, indicating the epitope of 1E10 is distinctly different from the rest of the antibodies. Based on this observation, MAb 1E10 was selected as the capturing antibody of the flagellin for pairwise comparisons of the other four antibodies. In contrast to the covalent immobilization of flagellin directly on the sensor surface, the immobilization of flagellin through the epitope of MAb 1E10 allowed other epitopes to be presented in a precise orientation and to be evaluated relative to the epitope of MAb 1E10. This approach also avoided the potential epitopes hindrance by covalent immobilization of the flagellin directly on the sensor surface and allowed for the comparison of more than two potential epitopes concurrently. In a previous report, liposomes were used to immobilize anti-flagellin antibodies for epitope mapping and kinetic analysis [44]. Our approach of immobilizing flagellin through an epitope using a capturing monoclonal antibody has not been reported elsewhere.

We have investigated five anti-flagellin antibodies to determine the epitope map using pairwise SPR assays. From the binding ratios (Table 2), it can be concluded that there are at least four epitopes recognized by the five different antibodies. A diagram of the four hypothesized epitopes of flagellin is presented in Figure 4. These conformational epitopes, centered by the epitope of MAb 1E10, are separated by a distance, and therefore, there is no interference with the binding of each other. 

Western blot of the trypsin-digested fragments suggested that these epitopes are located within a domain of 23 kDa on flagellin. Our results were coincident with the findings of other studies. Four distinct flagellin epitopes of *Salmonella* Muenchen were described using octameric peptides synthesized on polyethylene pins [48]. It has been proposed from other researchers [47] that there are three domains in flagellin, D1–D3, from the center of the filament axis outwards in the radial direction. The core domain (D1) is responsible for filament assembly and polymorphism. The middle domain (D2) may be related to the stability of the filament shape. The outer domains (D3) of adjacent subunits in a filament are not connected to each other. There are several lines of evidence showing that the exposed D3 domain contains the major epitopes of the flagellar antigen (H antigen). Other studies have provided evidence of the presence of discontinuous flagellin epitopes [8], and it was suggested that a minimal area, 86 amino acids for H:i and 102 amino acids for H:1,2, located in the central variable domain of each flagellin, was required for the binding of serotype-specific antibodies. In addition, there are some epitopes that may be shared by phase-1 and phase-2 flagellin. A previous study using ELISA-based epitope mapping has shown only one binding site on the recombinant flagellin of *S.* Typhimurium, where researchers investigated 6 antibodies, 3 for phase-1 and 3 for phase-2 flagellin [8]. 

Binding kinetics are crucial for the applications of these monoclonal antibodies in detection and subtyping studies. All antibodies significantly differ in terms of association rate, dissociation rate, and the equilibrium dissociation constant. We have compared several approaches to fitting the kinetic data; the best fittings were achieved using a 1:1 interaction model (Langmuir fit model). It should be noted that the presence of minor fragments in the flagellin preparations could affect the fitting and cause a small deviation from the 1:1 interaction model and may consequently contribute to slightly higher variations in the kinetic parameters. Nevertheless, this is the first study reporting on the kinetics of more than one anti-flagellin antibody with the captured flagellin of *S.* Typhimurium. 

## 5. Conclusions

We have demonstrated a real-time and label-free epitope mapping of *Salmonella* flagellin with SPR using a panel of anti-flagellin antibodies. With the epitope-captured-flagellin approach, four separated epitopes were identified by the five monoclonal antibodies. In addition, results from the Western blot of trypsin-digested fragments indicated all four epitopes are located within a 23 kDa domain of the flagellin. Collectively, epitope mapping and kinetics data obtained in this study are crucial for the further development and optimization of SPR and other immunoassays for the detection and subtyping of *Salmonella*. Based on the results of this study, SPR assays utilizing magnetic nanoparticle coupled monoclonal antibodies to enhance the sensitivity for the detection of *Salmonella* in food and environmental samples are being developed.

## Figures and Tables

**Figure 1 antibodies-08-00022-f001:**
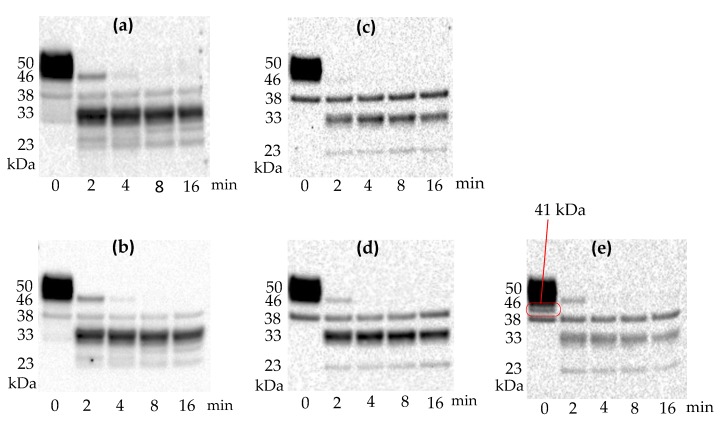
Western blot analysis of undigested flagellin and trypsin-digested flagellin fragments. Flagellin was incubated with trypsin at a protein ratio of 10:1 (*w*/*w*) for 2, 4, 8, and 16 min at room temperature. Five different anti-flagellin antibodies, (**a**) MAb 1C8, (**b**) MAb 7E3, (**c**) MAb 3H8, (**d**) MAb 5F11 and (**e**) MAb 1E10 were used to detect the flagellin and its fragments. Precision Plus Protein™ WesternC™ Standards were used for the molecular weight calibrations.

**Figure 2 antibodies-08-00022-f002:**
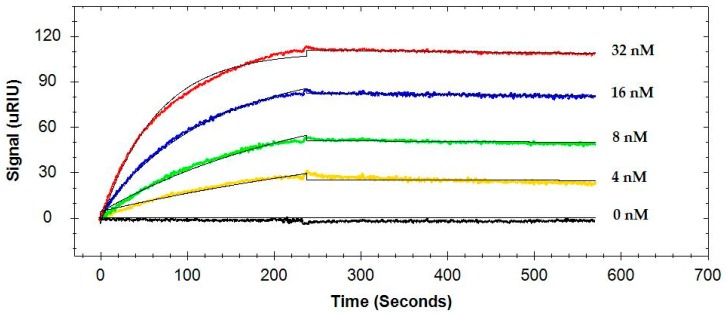
A representative sensorgram showing the interactions of four concentrations of the same antibody (MAb 7E3) to the flagellin captured on the sensor surface by immobilized MAb 1E10.

**Figure 3 antibodies-08-00022-f003:**
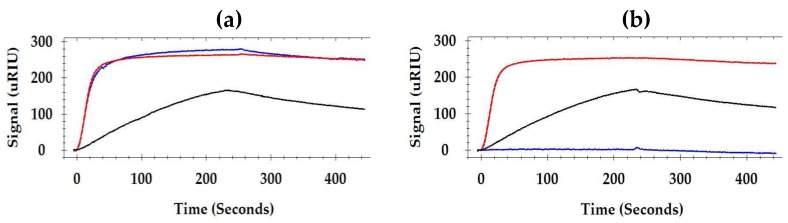
Representative SPR sensorgrams showing (**a**) non-inhibition and (**b**) inhibition of the binding of a pair of monoclonal antibodies; (**a**) Flagellin (black curve) was captured by MAb 1E10 immobilized on the surface. Immediately after the flagellin, MAb 1C8 (red curve) was injected and MAb 3H8 (blue curve) was followed after. Both MAbs (1C8 and 3H8) produced binding curves (non-inhibition); (**b**) Flagellin (black curve) was captured by MAb 1E10 immobilized on the surface. Immediately after the flagellin, MAb 1C8 (red curve) was injected and MAb 7E3 (blue curve) was followed after. MAb 7E3 failed to produce a binding curve (inhibition).

**Figure 4 antibodies-08-00022-f004:**
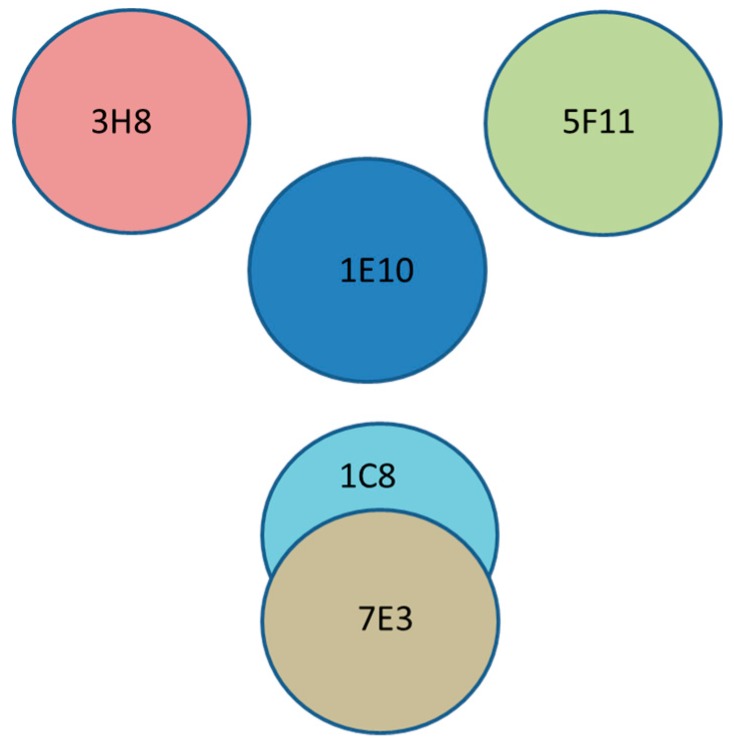
A diagram of conformational epitopes of flagellin centered by the epitope of MAb 1E10. Four distinct epitopes were identified using five anti-flagellin antibodies (MAb 3H8, MAb 5F11, MAb 1C8, MAb 7E3, and MAb 1E10). Overlapping circles denote common epitopes shared by MAb 1C8 and MAb 7E3.

**Table 1 antibodies-08-00022-t001:** Kinetic parameters of four monoclonal antibodies to S. Typhimurium flagellin.

Antibodies	*k_a_* (M^−1^s^−1^)	*k_d_* (s^−1^)	*K*_D_ (M)
MAb 1C8	6.43 × 10^5 b^ (± 5.23 × 10^4^)	7.09 × 10^−5 ab^ (± 3.55 × 10^−5^)	1.13 × 10^−10 ab^ (± 6.41 × 10^−11^)
MAb 7E3	4.88 × 10^5 b^ (± 9.26 × 10^4^)	1.03 × 10^−4 ab^ (± 1.53 × 10^−5^)	2.19 × 10^−10 ab^ (± 7.28 × 10^−11^)
MAb 3H8	6.68 × 10^5 b^ (± 6.36 × 10^3^)	2.17 × 10^−4 a^ (± 4.45 × 10^−5^)	3.25 × 10^−10 a^ (± 7.00 × 10^−11^)
MAb 5F11	1.57 × 10^6 a^ (± 9.19 × 10^4^)	3.98 × 10^−5 b^ (± 4.74 × 10^−5^)	2.63 × 10^−11 b^ (± 3.18 × 10^−11^)
Average	8.41 × 10^5^ (± 4.56 × 10^5^)	1.08 × 10^−4^ (± 7.69 × 10^−5^)	1.70 × 10^−10^ (± 1.29 × 10^−10^)
*p*-value	0.00	0.034	0.031

The numbers followed by different letters are significantly different; the values in parentheses are standard deviations.

**Table 2 antibodies-08-00022-t002:** Summary of the binding ratios of paired monoclonal antibodies on the immobilized flagellin.

First MAb	Second MAb			
	MAb 1C8	MAb 7E3	MAb 3H8	MAb 5F11
**MAb 1C8**	0.06 (± 0.01)	0.29 (± 0.02)	1.76 (± 0.48)	1.72 (± 0.07)
**MAb 7E3**	0.12 (± 0.00)	0.03 (± 0.01)	2.02 (± 0.01)	2.19 (± 0.23)
**MAb 3H8**	1.65 (± 0.11)	1.06 (± 0.01)	0.20 (± 0.04)	1.53 (± 0.08)
**MAb 5F11**	1.04 (± 0.06)	1.31 (± 0.02)	1.49 (± 0.43)	0.21 (± 0.03)

The binding ratio was obtained by dividing the maximum binding (Bmax) of second antibody to the Bmax of first antibody; the values in parentheses are standard deviations.

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
