# Peer review of "Kinetic Analysis and Epitope Mapping of Monoclonal Antibodies to Salmonella Typhimurium Flagellin Using a Surface Plasmon Resonance Biosensor"

_2073-4468, 2019, doi:10.3390/antib8010022_

Round 1

Reviewer 1 Report

72. affinity, dissociation constant, which is inverse of equilibrium dissociation constant

Section 2.2, please show the purity of the flagellin.

226-227, IE10 did not detect flagellin fragment at 33 kDa. It is wrong. IE10 also detected the band at 33 kDa (line 240-241 mentioned that all 5 antibodies detected falagellin at 38, 33 and 23 kDa).

255, 256, Table 1 and other places, rate constant should be the small/italic k, please change Ka to ka, Kd to kd, and dissociation constant KD to KD

Figure 2. The data fiting was not too good. Is the 1:1 ratio the only option? Please discuss the possible reason.

Table 1. The standard deviation is pretty high. It may be due to the not-so-good fit of data? Please discuss the possible reason.

Author Response

Response to Reviewer 1 Comments

Point 1: 72. Affinity, dissociation constant, which is inverse of equilibrium dissociation constant

Response 1: The sentence has been revised (L71-73).

Point 2: Section 2.2, please show the purity of the flagellin.

Response 2: Revised in Section 2.2 (L125-127), and described in Section 3.1 (L224-226).

Point 3: 226-227, IE10 did not detect flagellin fragment at 33 kDa. It is wrong. IE10 also detected the band at 33 kDa (line 240-241 mentioned that all 5 antibodies detected falagellin at 38, 33 and 23 kDa).

Response 3: The sentence has been revised (L230-232).

Point 4: 255, 256, Table 1 and other places, rate constant should be the small/italic k, please change Ka to ka, Kd to kd, and dissociation constant KD to KD

Response 4: Corrected accordingly

Point 5: Figure 2. The data fiting was not too good. Is the 1:1 ratio the only option? Please discuss the possible reason.

Response 5: Figure 2 has been replaced by a new sensorgram of 7E3 with better data fitting. A discussion has been added in Section 4 (L383-387).

Point 6: Table 1. The standard deviation is pretty high. It may be due to the not-so-good fit of data? Please discuss the possible reason.

Response 6: A discussion has been added in Section 4 (L383-387).                                        

Reviewer 2 Report

The authors present a comprehensive study describing the development of a SPR assay using the capture-based immobilization to determine relative binding sites of five monoclonal antibodies to S. Typhimurium flagellin.  Likewise, the binding kinetics of these monoclonal antibodies was investigated. 

Due to the significance of infections caused by Salmonella Typhimurium, the present study is expected to stimulate further research - for diagnosis of similar infectious microorganisms, too. Therefore, it should be of interest for the readership of Antibodies.

A few minor points may improve the manuscript:

Some sentences may be edited, e.g. in some cases an "a" should be added (such as in l. 321: "a" binding ratio)

Figure 2: The concentrations should be indicated in the figure.

A paragraph should be added providing details about further development and optimization of  biosensors and other immunoassays.

Author Response

Response to Reviewer 2 Comments

Point 1: Some sentences may be edited, e.g. in some cases an "a" should be added (such as in l. 321: "a" binding ratio)

Response 1: Edited accordingly

Point 2: Figure 2: The concentrations should be indicated in the figure.

Response 2: Concentrations have been indicated in Figure 2.

Point 3: A paragraph should be added providing details about further development and optimization of biosensors and other immunoassays.

Response 3: Further development of SPR assay has been described in Section 4 (L395-397).